# Leptin suppresses development of GLP-1 inputs to the paraventricular nucleus of the hypothalamus

Jessica E Biddinger[1]*, Roman M Lazarenko[1], Michael M Scott[2], Richard Simerly[1]*

[1]Department of Molecular Physiology and Biophysics, Vanderbilt University School of Medicine, Nashville, United States; [2]Department of Pharmacology, University of Virginia School of Medicine, Charlottesville, United States

**Abstract** The nucleus of the solitary tract (NTS) is critical for the central integration of signals from visceral organs and contains preproglucagon (PPG) neurons, which express leptin receptors in the mouse and send direct projections to the paraventricular nucleus of the hypothalamus (PVH). Here, we visualized projections of PPG neurons in leptin-deficient $Lep^{ob/ob}$ mice and found that projections from PPG neurons are elevated compared with controls, and PPG projections were normalized by targeted rescue of leptin receptors in $LepRb^{TB/TB}$ mice, which lack functional neuronal leptin receptors. Moreover, $Lep^{ob/ob}$ and $LepRb^{TB/TB}$ mice displayed increased levels of neuronal activation in the PVH following vagal stimulation, and whole-cell patch recordings of GLP-1 receptor-expressing PVH neurons revealed enhanced excitatory neurotransmission, suggesting that leptin acts cell autonomously to suppress representation of excitatory afferents from PPG neurons, thereby diminishing the impact of visceral sensory information on GLP-1 receptor-expressing neurons in the PVH.

*For correspondence:
jessica.biddinger@vanderbilt.edu
(JEB);
richard.simerly@vanderbilt.edu
(RS)

**Competing interests:** The authors declare that no competing interests exist.

## Introduction

Effective coordination of metabolic regulatory processes requires integration of hormonal signals with viscerosensory information conveyed from thoracic and abdominal viscera by the vagus nerve. Sensory components of the vagus innervate the digestive tract and provide afferent projections that target neurons in the nucleus of the solitary tract (NTS). This viscerosensory information is conveyed directly from the NTS to a variety of sites within the central nervous system, including discrete populations of hypothalamic neurons, where they regulate homeostatic processes (*Schwartz and Zeltser, 2013*; *Saper and Stornetta, 2015*).

The paraventricular nucleus of the hypothalamus (PVH) plays a particularly important role in coordination of neural signals controlling energy balance and receives direct inputs from the NTS (*Cunningham and Sawchenko, 1988*). Neurons in the PVH are activated by stimulation of vagal afferents that recruit viscerosensory ascending projections from the brainstem to the hypothalamus (*Grill and Hayes, 2012*). The PVH serves as a major site of neuroendocrine integration with a significant component of hormonally mediated information being relayed indirectly via neural connections. Notably, the PVH does not appear to be a major direct target for the fat-derived hormone leptin, yet significant numbers of PVH neurons are influenced by systemic changes in leptin levels (*Myers and Olson, 2012*; *Scott et al., 2009*). Leptin receptors are abundant, however, in regions that provide strong inputs to the PVH, such as the arcuate nucleus of the hypothalamus (ARH). Leptin also impacts the activity of NTS neurons, which appear to express the long form of the leptin receptor (LepRb), and this information may converge onto the PVH with leptin signals conveyed by other afferent populations (*Grill, 2010*; *Scott et al., 2011*).

Vagal inputs to the NTS are established before birth, but viscerosensory afferents to the hypothalamus appear to develop during early postnatal life. PVH neurons receive direct synaptic inputs from the NTS, and development of catecholaminergic projections from the brainstem to the PVH is particularly well characterized (*Rinaman et al., 2011*). Neurons that contain the peptide glucagon-like peptide-1 (GLP-1) are located primarily in the NTS (*Holt et al., 2018*) and appear to innervate the PVH during the first postnatal week. However, functional maturation of viscerosensory regulation of PVH neurons may emerge later in postnatal life; systemic injection of the hormone cholecystokinin (CCK) to newborn rats activates neurons in the NTS but does not appear to activate neurons in the PVH until adulthood (*Rinaman et al., 1994*).

GLP-1 is derived from its precursor, preproglucagon (PPG; *Bell et al., 1983*) and is not expressed in noradrenergic neurons but represents a distinct subset of glutamatergic neurons in the caudal NTS, and adjacent intermediate reticular nucleus (*Vrang et al., 2007*; *Llewellyn-Smith et al., 2011*; *Llewellyn-Smith et al., 2013*). Activation of PPG neurons, either chemogenetically or through vagal stimulation, suppresses food intake (*Turton et al., 1996*; *Williams et al., 2006 Gaykema et al., 2017*; *Liu et al., 2017*), and these neurons appear to be directly involved in central regulation of stress-induced hypophagia (*Maniscalco et al., 2015*; *Holt et al., 2018*; *Terrill et al., 2018*). Moreover, PPG neurons provide direct inputs to the PVH (*Sarkar et al., 2003*; *Vrang et al., 2007*; *Katsurada et al., 2014*) and LepRb are expressed on the majority of PPG neurons in the mouse NTS (*Huo et al., 2008*; *Garfield et al., 2012*), suggesting that information conveyed by PPG neurons may be modified by circulating leptin (*Hisadome et al., 2010*; *Clemmensen et al., 2017*).

Although it is clear that innervation of the PVH by neurons in the ARH is dependent on exposure to leptin during neonatal life (*Bouyer and Simerly, 2013*), it remains uncertain if leptin is required for normal development of other afferent neural systems. Because innervation of the PVH by GLP-1 axons occurs during a neonatal period of elevated leptin secretion (*Ahima and Flier, 2000*; *Elson and Simerly, 2015*), we used multiple molecular genetic approaches to test the hypothesis that leptin is required to support development of GLP-1 inputs to the PVH.

## Results

### Ontogeny of GLP-1 projections into the PVH

In neonatal mice, GLP-1 immunoreactive fibers were detected in the PVH as early as P6 (*Figure 1A*). Although GLP-1 fibers are visible at P6, their density is sparse at this age, and the preautonomic compartment of the PVH is nearly devoid of GLP-1 innervation. By P10, the density of GLP-1 immunoreactive fibers increases in neuroendocrine components of the PVH, with the dorsal region of the medial parvicellular part (PVHmpd) more heavily innervated than the ventral parvicellular (PVHmpv) compartment, and at this age, GLP-1 projections begin to extend into the lateral zone of the posterior magnocellular (PVHpml) compartment (*Figure 1B*). This overall pattern of GLP-1 fiber distribution was maintained through P16, although the density of GLP-1 fibers increased substantially in both the PVHmpd and PVHpml compartments (*Figure 1C*). By P24, the density of GLP-1 immunolabeled fibers was elevated further, with a higher density observed in the PVHmpd region, compared with that of the PVHpml (*Figure 1D*). At 2 months of age, the density and distribution of GLP-1 immunoreactive fibers in the PVH achieves levels that are similar to that of adult mice (*Figure 1E*). Throughout development, we observed GLP-1 immunoreactive fibers in the most rostral portions of the PVH (PVHap), or within the caudal preautonomic part of the PVH (PVHlp), but to a lesser degree than the neuroendocrine PVHmpd and PVHpml compartments.

### Leptin receptor-expressing neurons are responsive to leptin during early postnatal development and send direct projections to the PVH

To verify expression of LepRb in NTS neurons during postnatal development, we utilized *LepRb-Cre* mice to target expression of tdTomato (LepRb-Cre::tdTom mice). Moreover, these LepRb neurons appear to provide strong inputs to the PVH. To label terminal fields of NTS LepRb-expressing neurons, an AAV-encoding Cre-dependent EGFP virus was injected into the NTS of LepRb-Cre mice. The virus robustly labeled LepRb-expressing neurons in the NTS (*Figure 2F,G*) and revealed that the PVH receives direct projections from these neurons (*Figure 2H*). To confirm that LepRb are functional in NTS neurons during postnatal development we measured pSTAT3 immunolabeling in

**Figure 1.** Ontogeny of GLP-1 projections into the PVH. Maximum intensity projections of confocal images through the PVH of WT (**A–E**) and *Lep*$^{ob/ob}$ (**F–J**) mice illustrate the density of GLP-1 immunoreactive fibers (green) present at postnatal ages P6, P10, P16, P24, and P60. In WT mice, GLP-1 fibers first reach the PVH at P6 and continue to increase in density over time, reaching maturity by 2 months of age. In *Lep*$^{ob/ob}$ mice, a significant increase in GLP-1 fiber density emerged at P16 and was maintained into adulthood. Boxes indicate location of ROIs used for quantitative analysis. Student's t-test was used to test for significant differences between genotypes at each age examined. Asterisk denotes p-values<0.05. Abbreviations: 3V, third ventricle; mpd, dorsal zone of the medial parvicellular compartment of the PVH; pml, lateral zone of the posterior magnocellular compartment of the PVH. Error bars indicate mean ± SEM; circles represent individual values; N = 5 for each group. Scale bar is 50 μm.

leptin-treated LepRb-Cre::tdTom mice. Following i.p. leptin injection on P16, 85% of NTS LepRb-Cre::tdTom labeled neurons were positive for pSTAT3 immunoreactivity, in contrast to the lack of pSTAT3 immunolabeling in saline-injected controls (*Figure 2A–B,E*).

To visualize PPG neurons and their inputs into the PVH, a synaptophysin-tdTomato fusion protein was expressed under the *Gcg* promoter, resulting in GCG-Cre::SynTom mice. The distribution of SynTom-labeled neurons in GCG-Cre::SynTom mice was nearly identical to previous reports, with expression located exclusively to the medulla (*Jin et al., 1988*; *Vrang et al., 2007*; *Llewellyn-Smith et al., 2011*; *Llewellyn-Smith et al., 2013*; *Gu et al., 2013*; *Gaykema et al., 2017*). The majority of GCG-Cre::SynTom-labeled neurons were observed in the NTS, with a limited number of neurons (approximately 3–4 neurons per section) located in the adjacent intermediate reticular nucleus (IRT). We did not observe GCG-Cre::SynTom cell bodies outside of the medulla (*Figure 2—figure supplement 1*). In addition, GCG-Cre::SynTom mice were utilized to determine if PPG neurons are responsive to leptin during development. Following i.p. leptin injection at P16, 82% of Syn-Tom-labeled PPG neurons in the NTS displayed pSTAT3 immunoreactivity, as did the majority of PPG neurons located in the IRT, indicating that PPG neurons are responsive to leptin during postnatal development, which may impact targeting of GLP-1 projections to the hypothalamus (*Figure 2C–E*).



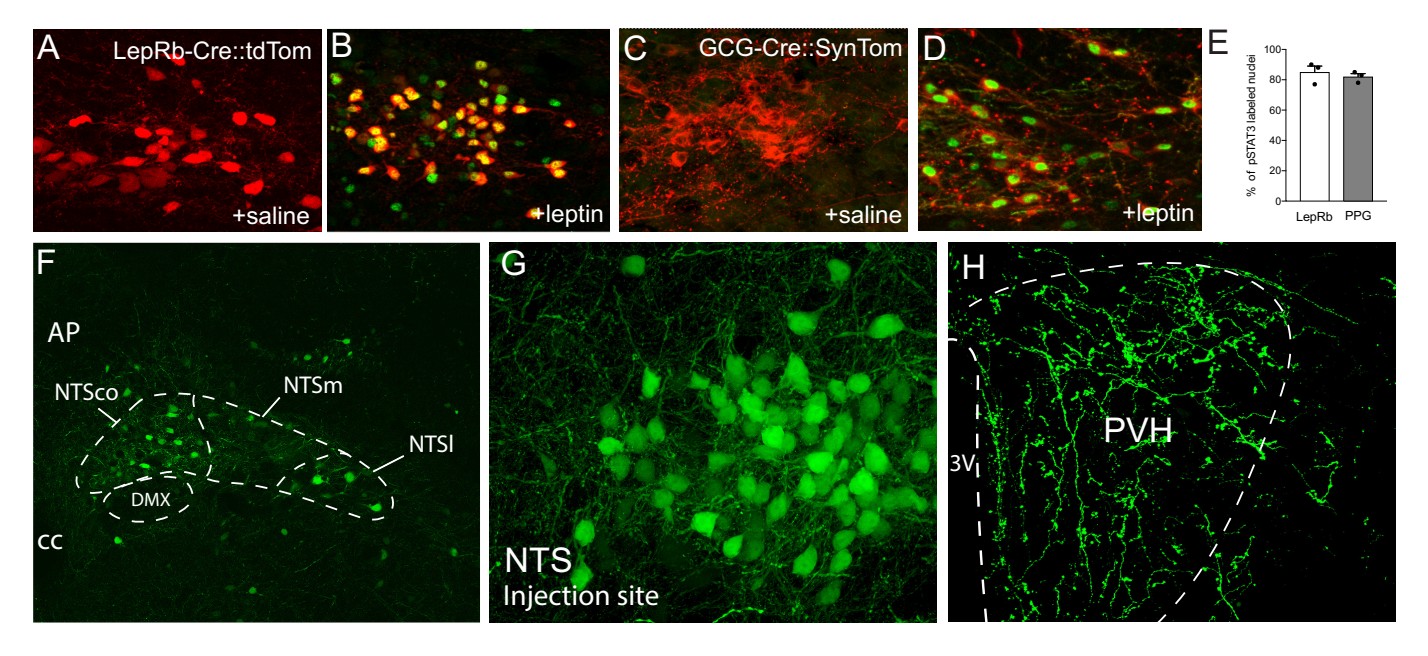

**Figure 2.** Leptin receptor-expressing neurons in the NTS are responsive to leptin during early postnatal development and project to the PVH. (A), (B) Representative images of pSTAT3 immunoreactivity (green) in LepRb-Cre::tdTom mice injected with leptin or saline control at P16. (C), (D) Representative images of pSTAT3 immunoreactivity in GCG-Cre::SynTom mice injected with leptin or saline control at P16. (E) Percentage of LepRb and PPG neurons in the NTS that are responsive to leptin. (F, G) LepRb-expressing neurons are visualized by injection of AAV-EGFP virus into the NTS of a LepRb-Cre mouse. (H) Densely labeled terminals in the PVH from LepRb-expressing neurons in the NTS. N = 3 for each group. Abbreviations: 3V, third ventricle; AP, area postrema; DMX, dorsal motor nucleus of the vagus; NTSco, commissural nucleus of the solitary tract; NTSm, medial nucleus of the solitary tract; NTSl, lateral nucleus of the solitary tract; cc, central canal.

The online version of this article includes the following figure supplement(s) for figure 2:

**Figure supplement 1.** Distribution of GCG-Cre::SynTom expression in the hindbrain and hypothalamus.

## GLP-1 projections to the PVH are increased in leptin-deficient mice

To determine if leptin is required for normal development of GLP-1 projections to the hypothalamus, the density of GLP-1 immunoreactive inputs to the PVH was evaluated in leptin-deficient mice. Although immunolabeled GLP-1 fibers were clearly apparent in the PVH by P6 and increased in density over time, no statistically significant differences in their density were detected in $Lep^{ob/ob}$ mice compared to WT controls, in either the PVHmpd or the PVHpml at either P6 or P10 (*Figure 1*). However, by P16, the density of GLP-1 immunoreactive fibers in the PVHmpd of $Lep^{ob/ob}$ mice was 32% greater than that of WT mice (*Figure 1C,H,M*; p=0.037). At this age, no significant differences were detected in the density of GLP-1 immunoreactive fibers in the PVHpml (*Figure 1C,H,M*). At P24, the density of labeled fibers increased by 46.3% in the PVHmpd, and by 34.2% in the PVHpml, of $Lep^{ob/ob}$ mice, compared with WT controls (*Figure 1D,I,N*; p=0.0305 and p=0.0493, respectively). The increase in GLP-1 fiber density in the PVH of $Lep^{ob/ob}$ mice was maintained into adulthood (P60) in both the PVHmpd and PVHpml (70.2 and 61.3% higher relative to WT mice, respectively; *Figure 1E, J,O*; p=0.0061 and p=0.0143, respectively). Because expression of GLP-1 immunoreactivity may change in the absence of leptin, we confirmed these findings by using leptin-deficient GCG-Cre::SynTom (GCG-Cre::SynTom::$Lep^{ob/ob}$) mice to visualize axons derived from PPG neurons. In alignment with our immunohistochemical results, at P16 the density of SynTom-labeled inputs was 37.4% higher in the PVHmpd of GCG-Cre::SynTom::$Lep^{ob/ob}$ mice, and 33.8% higher in the PVHpml, compared with GCG-Cre::SynTom::WT mice (*Figure 3A–C*; p=0.0308 and p=0.0408, respectively). Furthermore, the increase in the density of GLP-1 axonal labeling in the PVH was maintained into adulthood: GCG-Cre::SynTom::$Lep^{ob/ob}$ mice displayed a 50% increase in the density of GCG-Cre::SynTom inputs to the PVHmpd, and a 40.2% increase in the PVHpml, compared with that of GCG-



**Figure 3.** Axonal labeling of projections from PPG neurons to the PVH are increased in leptin-deficient mice. Genetic targeting of SynTom fluorescence to axons derived from PPG neurons revealed a significant increase in the density of GLP-1 projections to the PVHmpd and PVHpml in *Lep^ob/ob* mice. Quantitative analysis confirmed that fiber densities in both the mpd and pml (ROI location denoted by boxes) were apparent by P16 (**A–C**), and were maintained into adulthood (**D–F**). Leptin does not impact the density of GLP-1 inputs to the ARH, as no changes in GCG-Cre::SynTom fluorescence were identified in WT or *Lep^ob/ob* mice (**G–I**). No difference in the number of PPG neurons was detected between groups of adult mice (**J–L**). Error bars indicate mean ± SEM; circles represent individual values. Student's t-test was used to test for significant differences between genotypes at each age

*Figure 3 continued on next page*

*Figure 3 continued*

examined. Asterisk denotes p-values<0.05. Images are maximum intensity projections from confocal image stacks taken through 30 μm-thick sections. P16 GCG-Cre::SynTom::WT N = 3; P16 GCG-Cre::SynTom::Lep^{ob/ob}N = 4; P60 GCG-Cre::SynTom::WT N = 4; P60 GCG-Cre::SynTom::Lep^{ob/ob}N = 6, PPG neurons: N = 6 for each group. Abbreviations: 3V, third ventricle; mpd, dorsal zone of the medial parvicellular compartment of the PVH; pml, lateral zone of the posterior magnocellular compartment of the PVH; ARH, arcuate nucleus of the hypothalamus; AP, area postrema; DMX, dorsal motor nucleus of the vagus; NTS, nucleus of the solitary tract; cc, central canal. Scale bar is 50 μm.

Cre::SynTom::WT mice at P60 (*Figure 3D–F*; p=0.0089 and p=0.0045 respectively). Leptin does not affect GLP-1 axonal inputs to the ARH, as no changes were detected in GLP-1 axon density between WT and GCG-Cre::SynTom::Lep^{ob/ob}mice, indicating that leptin does not appear to uniformly impact the projection fields of PPG neurons (*Figure 3G–I*). Additionally, the increase in the density of Syn-Tom inputs to the PVH of GCG-Cre-SynTom::Lep^{ob/ob} mice did not appear to be associated with any changes in the density or distribution of PPG neurons (*Figure 3J–L*), suggesting that leptin impacts targeting of axons to the PVH, but does not alter PPG neuron number.

## Target-specific enhancement of GLP-1 inputs to CRH neurons in leptin-deficient mice

To determine if targeting of GLP-1 inputs to specific subpopulations of PVH neurons is impacted by leptin, we evaluated the density of GCG-Cre::SynTom terminals onto corticotrophin-releasing hormone (CRH) and Oxytocin neurons in neuroendocrine compartments of the PVH in leptin-deficient mice. CRH and Oxytocin neurons in the PVH were visualized using immunohistochemistry in GCG-Cre::SynTom::Lep^{ob/ob} mice and GCG-Cre::SynTom::WT controls. In WT mice, GLP-1 inputs to CRH neurons appeared more numerous than those to Oxytocin neurons: the density of GCG-Cre::Syn-Tom-labeled inputs to CRH neurons were approximately 50% greater than those to Oxytocin neurons (*Figure 4A,D*). Moreover, leptin appeared to preferentially impact the density of GLP-1 inputs onto CRH neurons, as we observed a greater number of GCG-Cre::SynTom-labeled terminals in close association with labeled CRH neurons in adult GCG-Cre::SynTom::Lep^{ob/ob} mice than in GCG-Cre::SynTom::WT controls (*Figure 4D–F*; p=0.0017). However, leptin does not appear to alter the density of GLP-1 projections to oxytocin neurons because we did not detect a significant change in the mean number of GCG-Cre::SynTom terminals onto Oxytocin neurons (*Figure 4A–C*).

## Leptin-deficient mice have dysregulated viscerosensory transmission and hyper-representation of glutamatergic inputs to GLP-1 R neurons in the PVH

To determine if leptin alters the ability of PPG neurons to convey visceral sensory information to the hypothalamus, we injected CCK i.p. to activate vagal afferents and measured the density of Fos immunostained nuclei in the NTS and PVH of GCG-Cre::SynTom::Lep^{ob/ob} mice and WT controls. Consistent with previous findings in rats (*Rinaman, 1999*; *Maniscalco and Rinaman, 2013*), CCK administration results in an induction of Fos immunoreactivity in the majority of PPG neurons. Injection of CCK to GCG-Cre::SynTom::WT mice caused an induction of Fos immunoreactivity in 95% of labeled PPG neurons (located in the NTS and IRT), compared with saline-injected controls (*Figure 5* insets A, B, D). Similar levels of Fos labeling were observed in PPG neurons of GCG-Cre::SynTom::WT and GCG-Cre::SynTom::Lep^{ob/ob} mice following CCK injection (*Figure 5C* inset,D). However, CCK administration resulted in a significantly higher number of Fos immunolabeled nuclei in the PVHmpd of GCG-Cre::SynTom::WT mice than in saline-injected GCG-Cre::SynTom::WT controls (*Figure 5A–D*; p<0.0002). Further, injection of CCK resulted in a 64% increase in the number of Fos immunolabeled nuclei in the PVHmpd of GCG-Cre::SynTom::Lep^{ob/ob} mice, compared with CCK-injected GCG-Cre::SynTom::WT mice (*Figure 5A–D*; p<0.0001). To test whether leptin alters the activity of postsynaptic neurons in the PVH that receive GLP-1 inputs, we used whole-cell patch-clamp electrophysiology to record miniature excitatory postsynaptic currents (mEPSCs) from PVH neurons that express GLP-1 receptors (GLP-1 R), visualized by crossing *Glp1r-Cre* mice with the Cre-dependent fluorescent reporter tdTomato in leptin-deficient mice (GLP-1 R-Cre::tdTom::Lep^{ob/ob} mice). Leptin did not impact the kinetics of glutamate signaling, as there were no differences measured in the rise and decay times of averaged mEPSC events normalized to peak current in GLP-1 R-Cre::tdTom::WT controls and GLP-1 R-Cre::tdTom::Lep^{ob/ob} mice (*Figure 6G*). Consistent with the

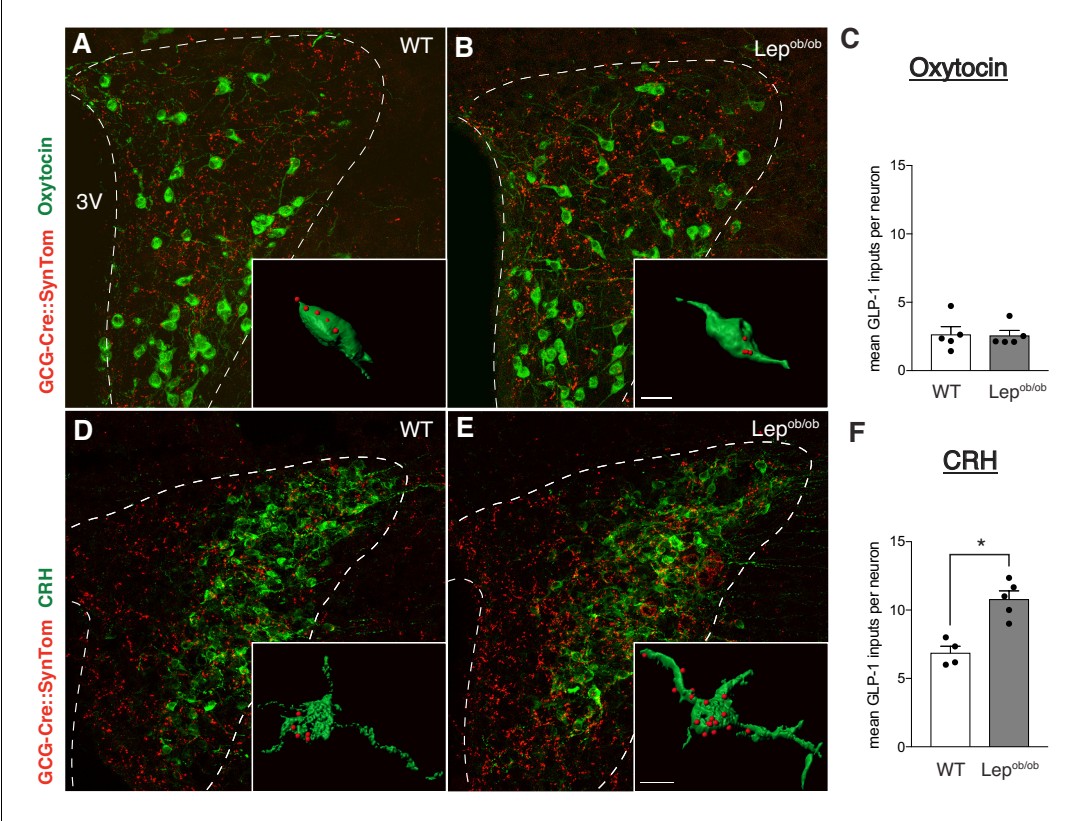

**Figure 4.** Target-specific enhancement of GLP-1 inputs to CRH neurons in leptin-deficient mice. Images (maximum intensity projections of confocal images) showing GLP-1 axonal labeling (red) onto Oxytocin- (A,B), or CRH-immunoreactive (D,E) neurons (green) in the PVH of GCG-Cre::SynTom::WT and GCG-Cre::SynTom::Lep^ob/ob mice. Insets illustrate 3D reconstructions of GLP-1 inputs onto Oxytocin and CRH neurons in GCG-Cre::SynTom::WT and GCG-Cre::SynTom::Lep^ob/ob mice, respectively. Quantitative analysis of GLP-1 inputs onto PVH neurons revealed a significant increase onto CRH (F), but not Oxytocin (C) neurons in GCG-Cre::SynTom::Lep^ob/ob mice, compared with that of GCG-Cre::SynTom::WT mice. Error bars indicate mean ± SEM; circles represent individual values. Student's t-test was used to test for significant differences between genotypes for each cell type (CRH or Oxytocin) examined. Asterisk denotes p-values<0.05. Oxytocin: GCG-Cre::SynTom::WT N = 5; GCG-Cre::SynTom::Lep^ob/ob N = 5; CRH: GCG-Cre:: SynTom::WT N = 4 GCG-Cre::SynTom::Lep^ob/ob N = 5. Scale bar is 50 µm, inset scale bar is 10 µm.

increased levels of Fos immunoreactivity detected in the PVH of leptin-deficient mice, we observed a significant and selective increase in the frequency of mEPSCs in GLP-1 R neurons in slices isolated from GLP-1 R-Cre::tdTom::Lep^ob/ob mice, compared with mEPSCs recorded from labeled neurons in GLP-1 R-Cre::tdTom::WT controls (*Figure 6H*). The increase in mEPSC frequency observed in GLP-1 R-Cre::tdTom::Lep^ob/ob mice appeared to be due to a 41% decrease in inter-event intervals in leptin-deficient mice (*Figure 6I*; p<0.0001). Because PPG neurons are glutamatergic, these results are consistent with enhanced excitatory neurotransmission in GLP-1 R neurons of leptin-deficient mice. There was no difference in mEPSC amplitude between GLP-1 R-Cre::tdTom::Lep^ob/ob mice and WT controls (*Figure 6J*), which indicates that leptin does not directly alter the sensitivity of postsynaptic GLP-1 R neurons in the PVH to glutamate.

## Cell-autonomous action of leptin in the NTS specifies the density and activity of GLP-1 inputs to PVH neurons

Leptin receptors are expressed on nearly all PPG neurons, but they are also expressed in a number of regions that provide afferents to the NTS and PVH. Therefore, we used a combined loss of function/gain of function molecular genetic approach to examine the site of action for the developmental regulation of PPG inputs to the PVH by leptin. Immunohistochemistry was utilized to visualize GLP-1 projections in functionally leptin-receptor null *LepRb^TB/TB* mice, which effectively lack expression of



**Figure 5.** Viscerosensory transmission to the PVH is disrupted in leptin-deficient mice. Compared with saline-injected controls (**A**), i.p. injection of CCK increased the number of Fos immunoreactive nuclei (green) in the PVH (**B,C**), and within PPG-labeled neurons (red) in the NTS (insets) of adult mice. GCG-Cre::SynTom::Lep$^{ob/ob}$ mice showed a significant increase in the number of Fos-labeled nuclei in the PVH (**C**), compared with that observed in CCK injected WT mice (**B**), or saline-injected controls (**A**). Images are maximum intensity projections from confocal image stacks. Quantitative comparison between the density of Fos labeling in the PVH of WT mice treated with saline (white bars), WT mice treated with CCK (gray bars) or Lep$^{ob/ob}$ mice treated with CCK (black bars) confirmed the apparent induction in Fos labeling in Lep$^{ob/ob}$ mice (**D**). The percentage of PPG neurons in the NTS labeled with Fos in each group is shown at right. GCG-Cre::SynTom::WT + saline N = 5, GCG-Cre::SynTom::WT + CCK N = 5, GCG-Cre::SynTom::Lep$^{ob/ob}$ + CCK N = 7. Representative DIC image of identified PVH neuron used for whole-cell patch recording from acute brain slice derived from adult GLP-1 R-Cre::tdTom mouse (**E**). Fluorescence image from the same field of view as in E, illustrating tdTomato-labeling (**F**). Average rise and decay times of mEPSC events normalized to peak current did not differ between WT and Lep$^{ob/ob}$ mice (**G**). Representative traces of mEPSC recordings from a GLP1-R neuron in the PVH of WT (black traces) and Lep$^{ob/ob}$ mice (red traces) mice (**H**). Cumulative mEPSC inter-event interval (**I**) and amplitude (**J**) with bar graphs showing the averaged data for 14 neurons. Mean mEPSC frequency was significantly increased in Lep$^{ob/ob}$ mice. GLP-1 R-Cre::tdTom::WT N = 4 animals, 14 neurons; GLP-1 R-Cre::tdTom::Lep$^{ob/ob}$N = 6 animals, 14 neurons. Student's t-test was used to test for significant differences between treatment group. Asterisk denotes p-values<0.05. Error bars indicate mean ± SEM; circles represent individual values.

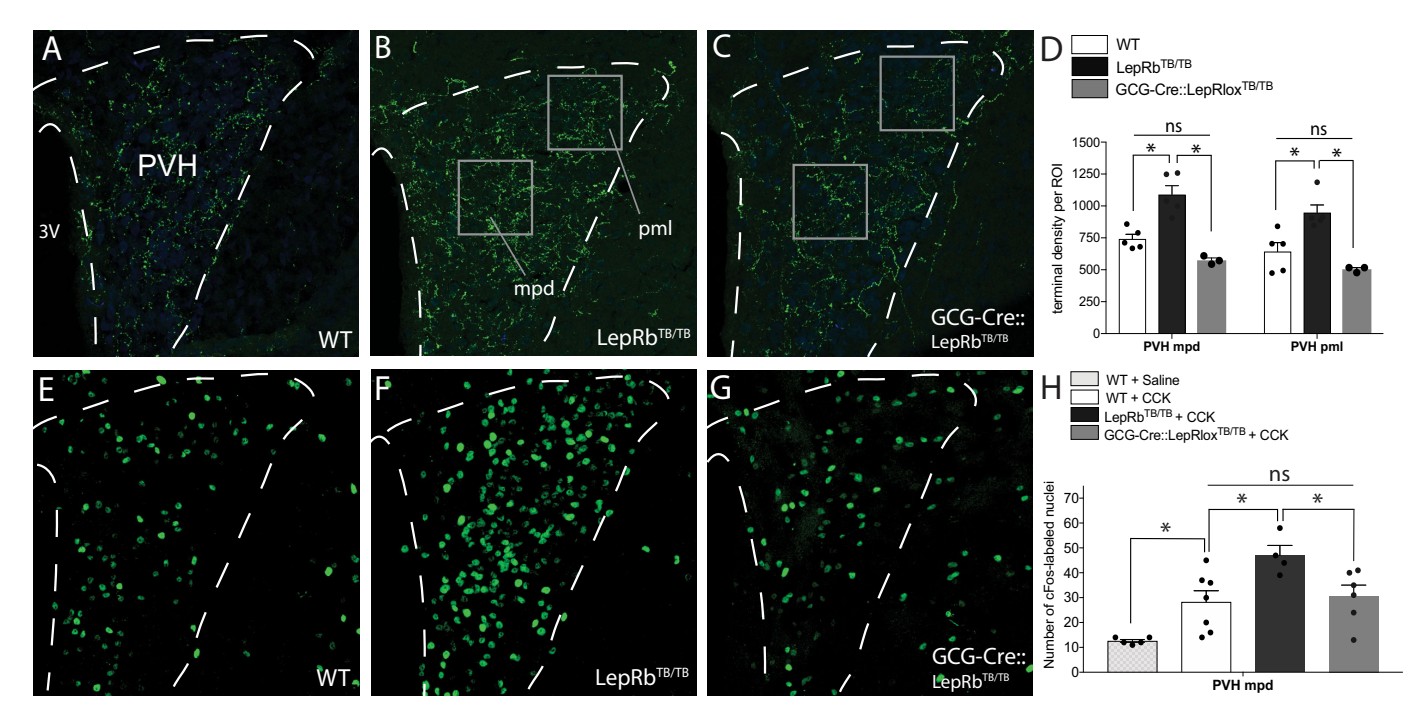

**Figure 6.** Cell-autonomous action of leptin in the NTS specifies the density and activity of GLP-1 inputs to PVH neurons. Compared with WT mice (A), mice functionally null for LepRb (LepRb^TB/TB mice) exhibited increased GLP-1-immunoreactive fiber density in both the mpd and pml compartments of the PVH (B), similar to that observed in Lep^ob/ob mice. Restoration of LepRb expression in PPG neurons (GCG-Cre::LepRb^TB/TB mice) reduced the density of GLP-1-immunoreactive fibers in the PVHmpd and PVHpml (C). Boxes indicate locations of the ROIs used for quantitative comparisons between groups (D). Activation of vagal afferents by CCK injection resulted in an increased number of Fos-labeled nuclei in the PVHmpd of LepRb^TB/TB mice (F) compared with WT mice that received either saline or CCK (E, H). Restoration of LepRb in PPG neurons (GCG-Cre::LepRb^TB/TB mice) normalized the number of Fos immunoreactive nuclei similar to those observed in the PVHmpd of WT mice (G, H). Abbreviations, 3V, third ventricle; mpd, dorsal zone of the medial parvicellular compartment of the PVH; pml, lateral zone of the posterior magnocellular compartment of the PVH. Error bars indicate mean ± SEM; circles represent individual values. One-way ANOVA was used to test for significant differences between genotypes, followed by post-hoc comparisons. Asterisk denotes p-values<0.05. Sample sizes for GLP-1 immunolabeling: WT N = 6, LepRb^TB/TB N = 5, GCG-Cre::LepRb^TB/TB N = 3. Sample sizes for CCK-induced Fos immunoreactivity: WT+ Saline N = 5, WT+ CCK N = 7, LepRb^TB/TB + CCK N = 4, GCG-Cre:: LepRb^TB/TB + CCK N = 6.

LepRb throughout the CNS (LepRb^TB/TB mice) due to the insertion of a *LoxP*-flanked transcription-blocking cassette in the *Lepr* gene (*Berglund et al., 2012*). The density of GLP-1 immunolabeled inputs to the PVH of LepRb^TB/TB mice was approximately 47% higher than in WT mice (*Figure 6A,B, D*; p=0.0022). Furthermore, the increase in the density of immunolabeled GLP-1 inputs observed in LepRb^TB/TB mice was almost identical to that observed in leptin-deficient mice when compared with WT mice. To restore LepRb signaling specifically in PPG neurons, LepRb^TB/TB mice were crossed with GCG-Cre mice (resulting in GCG-Cre::LepRb^TB/TB mice) and the density of GLP-1 immunoreactive fibers was measured in the PVH. Targeted expression of LepRb in PPG neurons appeared to restore the density of GLP-1 fibers in both the PVHmpd and PVHpml to levels that were similar to those observed in WT controls (*Figure 6C,D*), suggesting that LepRb functions cell autonomously to reduce targeting of GLP-1 projections to the PVH. Leptin does not alter PPG neuron number in either GCG-Cre::LepRb^TB/TB or LepRb^TB/TB mice, a finding nearly identical to that of the density and distribution of PPG neuron number in leptin-deficient mice (WT: 19.67 ± 0.8; LepRb^TB/TB: 20.02 ± 0.2; GCG-Cre::LepRb^TB/TB: 19.78 ± 0.7 PPG neurons per section). To assess whether manipulation of LepRb on PPG neurons impacts viscerosensory transmission from the gut to the brain, vagal afferents were stimulated by i.p. injection of CCK and the number of Fos immunoreactive nuclei in the PVH of WT, LepRb^TB/TB, and GCG-Cre::LepRb^TB/TB mice was quantified. Injection of CCK in WT mice resulted in a significant increase in the number of Fos immunoreactive nuclei in the PVH compared with WT mice that received saline injections (*Figure 6E,H*; p=0.0156). Following CCK

injection, a significant 65% increase in immunolabeled Fos nuclei was detected in global leptin-receptor null LepRb[TB/TB] mice compared with WT mice (*Figure 6E,F,H*; p=0.0213), consistent with our observations in Lep[ob/ob] mice, which suggests leptin specifies GLP-1 projections into the PVH through LepRb signaling. Further, restoration of LepRb signaling specifically in PPG neurons on an otherwise functionally LepRb null background normalized densities of Fos-labeled nuclei in the PVH of GCG-Cre::LepRb[TB/TB] mice, compared to those observed in WT mice (*Figure 6E,G,H*; p=0.4031, ns). This finding suggests that the activity of GLP-1 neural circuitry is regulated by LepRb signaling in a cell-autonomous manner. Taken together with results obtained in leptin-deficient Lep[ob/ob] mice (see *Figure 5*), our observations in leptin receptor-deficient LepRb[TB/TB] mice suggest that leptin alters transmission of viscerosensory information primarily by acting directly on PPG neurons during development of their ascending projections.

### Targeted expression of LepRb on PPG neurons does not normalize feeding behavior and anxiety-like behaviors observed in leptin receptor-deficient mice

In order to determine the role of LepRb on PPG neurons in regulating food intake and anxiety-like behaviors, we analyzed meal patterns, performance on the elevated zero maze test and novelty-suppressed feeding test, and measured blood serum corticosterone levels in GCG-Cre::LepRb[TB/TB] mice with targeted expression of LepRb on PPG neurons, global LepRb-deficient (LepRb[TB/TB] mice), and WT controls. Leptin-receptor null (LepRb[TB/TB] mice) demonstrated increased body weight and food intake compared with WT mice (*Figure 7A,B*; p=<0.0001 and p=0.0004, respectively), consistent with previous reports (*Berglund et al., 2012*). However, targeted expression of LepRb on PPG neurons did not normalize body weight or food intake to wild-type levels, and these GCG-Cre::LepRb[TB/TB] mice remained obese and ate significantly more food than WT mice (*Figure 7A,B*; p=0.0046). Furthermore, targeted expression of LepRb on PPG neurons did not normalize any meal pattern parameters analyzed to those characteristic of wild-type mice (*Figure 7C,D*). GCG-Cre::LepRb[TB/TB] mice displayed no difference in meal number or meal size compared to LepRb[TB/TB] mice, although both of these groups displayed a significant decrease in meal number and increase in meal size compared with WT mice (*Figure 7C,D*; p=0.0013 and p=0.0004, respectively). Because GLP-1 is involved in the regulation of stress and anxiety (*Tauchi et al., 2008*; *Terrill et al., 2018*), we also tested whether re-expression of LepRb on PPG neurons regulates anxiety-like behavior by performance on the elevated zero maze. As expected, LepRb[TB/TB] mice displayed increased anxiety-like behaviors by increased percent of time spent in the closed arm (*Figure 7F*; p=0.0252), decreased number of transitions into the open zone (*Figure 7G*; p=0.0013) and significantly decreased total distance traveled compared with WT mice (*Figure 7H*; p=0.0498). However, targeted expression of LepRb on PPG neurons did not normalize any aspects of performance on the elevated zero maze, with GCG-Cre::LepRb[TB/TB] mice displaying no differences compared to LepRb[TB/TB] mice (*Figure 7F–H*). In addition, blood levels of serum corticosterone were elevated in LepRb[TB/TB] mice compared with WT mice (p=0.0329), but corticosterone levels in GCG-Cre::LepRb[TB/TB] mice did not differ from those of LepRb[TB/TB] mice (*Figure 7I*). The novelty-suppressed feeding test was used to assess ingestive behaviors in an aversive environment and the latency to initiate a feeding bout and amount consumed was measured. Consistent with the elevated anxiety-like behavior displayed by LepRb[TB/TB] mice in the elevated zero maze, these animals also showed significantly longer time to initiate feeding (p<0.0001) and decreased total distance traveled (p=0.0014) compared with WT mice, although total amount of food consumed was not different between the groups (*Figure 7J–L*). Restoration of LepRb on PPG neurons did not normalize performance on the novelty-suppressed feeding test, as GCG-Cre::LepRb[TB/TB] mice did not differ in their responses compared to LepRb[TB/TB] mice (*Figure 7J–L*).

## Discussion

Accurate central representation of viscerosensory information depends on innervation of NTS neurons, which provide direct neural projections to forebrain sites that mediate multiple aspects of physiological homeostasis. Visceral sensory signals are conveyed from the gut to the brain primarily through vagal signaling. An intact vagus nerve is required for normal regulation of feeding behavior (*Schwartz et al., 1999*; *Fox et al., 2013*) and disrupted NTS to hypothalamus signaling can impair

**Figure 7.** Leptin's action on PPG neurons is not sufficient to restore food intake and anxiety-like behavior in leptin receptor-deficient mice. Compared with WT controls, LepRb[TB/TB] mice displayed increased body weight (**A**), increased total pellets consumed (**B**), decreased meal number (**C**) and increased meal size (**D**). Restoration of LepRb expression in PPG neurons (GCG-Cre::LepRb[TB/TB] mice) did not lead to normalization of any meal pattern parameters measured (**A–D**). Compared with WT controls, LepRb[TB/TB] mice demonstrated increased percent of time spent in the closed zone of the elevated zero maze (**F**), decreased number of transitions (**G**), decreased total distance traveled (**H**), and increased corticosterone levels (**I**). Restoration of LepRb expression in PPG neurons (GCG-Cre::LepRb[TB/TB] mice) did not lead to normalization of any aspects of the elevated zero maze (**F–H**) or blood corticosterone levels (**I**). Compared with WT controls, LepRb[TB/TB] mice displayed increased latency to feed (**J**), while there were no changes in total amount consumed (**K**), and decreased total distance traveled (**L**) in the novelty-suppressed feeding test. Restoration of LepRb expression in PPG neurons (GCG-Cre::LepRb[TB/TB] mice) did not lead to normalization of any parameters assessed in the novelty-suppressed feeding test (**J–L**). Error bars indicate mean ± SEM; circles represent individual values. One-way ANOVA was used to test for significant differences between genotypes, followed by post-hoc comparisons. Asterisk denotes p-values<0.05. Meal patterns: WT N = 6, LepRb[TB/TB]N = 5, GCG-Cre::LepRb[TB/TB]N = 5; Anxiety-like behavior: WT N = 10, LepRb[TB/TB]N = 5, GCG-Cre::LepRb[TB/TB]N = 6; Novelty-suppressed feeding test: WT N = 10, LepRb[TB/TB]N = 5, GCG-Cre::LepRb[TB/TB]N = 6.

normal patterns of food intake and body weight regulation (*D'Agostino et al., 2016*; *Liu et al., 2017*). PPG neurons convey viscerosensory information to the hypothalamus and the present study suggests that leptin is required for normal development of GLP-1 projections to the PVH. However, in contrast to its growth-promoting action in the hypothalamus (*Bouret et al., 2004a*) our findings indicate that leptin suppresses development of GLP-1 projections to the PVH. This novel finding significantly extends our understanding of how leptin may function to sculpt the organization of neural circuitry underlying regulation of energy balance.

The development of PVH inputs occurs primarily during the first three weeks of life. Axon tracing studies indicate that ARH fibers first reach the PVH at P10 and achieve an adult distribution by P18

(**Bouret et al., 2004a**). Catecholamine inputs into the PVH from the caudal brainstem are also established during postnatal life; dopamine-β-hydroxylase (DBH) fibers are present in the PVH at P1, but at a low density, and do not reach adult-like levels until P21. In contrast, phenylethanolamine-N-methyltransferase (PNMT) fibers are observed at a high density in the PVH at P1, and gradually decrease to adult-like levels at P21 (**Rinaman, 2001**). We first observed GLP-1 immunoreactive fibers in the PVH at P6, and the density of fibers gradually increased postnatally. This corresponds to a time when pups begin to leave the nest and initiate independent ingestion to enhance growth and survival (**Thiels et al., 1990**).

The density of GLP-1 inputs to neuroendocrine compartments of the PVH are significantly increased in leptin-deficient mice, demonstrating that leptin exposure during postnatal life is required to establish normal densities of GLP-1 inputs to PVH neurons. These findings are surprising, given that leptin's previously identified role in development was to promote axon outgrowth. Direct application of leptin to ARH explants results in enhanced axon outgrowth, and leptin-deficient mice show decreased AgRP innervation of the PVH (**Bouret et al., 2004b**), suggesting a growth-promoting neurotrophic role for leptin. In contrast to these findings, our results represent the first example of leptin suppressing axon outgrowth in the central nervous system during development. Suppression of axon outgrowth has been demonstrated in other sensory systems. For example, mice with mutations in brain-derived neurotrophic factor (BDNF) display increased sympathetic innervation of hair follicles and enhanced innervation of Merkel cells in the skin (**Fundin et al., 1997**; **Rice et al., 1998**). Alternatively, malfunction of axon pruning during development may be responsible for the increase in GLP-1 inputs to the PVH in leptin-deficient mice. Early in life, an overabundance of axon targeting is often observed, which is then refined to match neuronal requirements specified by activity or chemical signals. Strong, active inputs to a target region tend to maintain their connections, while weaker inputs are more likely to be pruned. Notably, axons that travel long distances are often pruned, and leptin may play a role in reducing the density of projections from the NTS to the PVH. Regulation of cell number may also be a plausible mechanism for regulating the density of inputs to the PVH from NTS. However, this seems an unlikely explanation for our results, because we did not observe a change in the number of PPG neurons in the NTS of either Lep$^{ob/ob}$ or Lep$^{TB/TB}$ mice. In adult mice, administration of leptin leads to an increase in the activity of PPG neurons (**Hisadome et al., 2010**), which would appear to be at odds with our results. However, leptin also has divergent actions on AgRP neurons during development and adulthood; during development leptin functions as a neurotrophic factor, promoting axon outgrowth from AgRP neurons, while in adulthood it functions to reflect the levels of body fat stores and inhibits the activity of AgRP neurons. In our study, leptin similarly functions as a neurotrophic factor that is required for the organization of GLP-1 inputs to the PVH early in life, but in contrast to leptin's growth-promoting role on AgRP neurons, leptin appears to suppress axon outgrowth from PPG neurons.

Leptin does not appear to suppress GLP-1 inputs uniformly to all PVH neurons. In the absence of leptin, increased innervation of CRH neurons by PPG-derived axons were particularly pronounced, while there was no change in the density of PPG inputs onto Oxytocin neurons. A similar cell-type specificity was observed for the developmental action of leptin on AgRP projections to the PVH. Treatment of neonatal Lep$^{ob/ob}$ mice with leptin restored AgRP inputs onto PVH preautonomic neurons, yet leptin treatment did not restore AgRP inputs onto neuroendocrine components of the PVH (**Bouyer and Simerly, 2013**). One possible explanation for the target-specific effects of leptin on GLP-1 projections to Oxytocin and CRH neurons is that each cell type may be innervated by different subpopulations of PPG neurons. However, at the current time there is no data suggesting PPG neurons are topographically organized into anatomically or functionally distinct subpopulations, and instead are currently considered to be a homogeneous cell group. Retrograde tracing studies have demonstrated that PPG neurons in the NTS and IRT project to the same terminal sites (**Rinaman, 1999**; **Vrang et al., 2007**; **Maniscalco and Rinaman, 2013**; **Cork et al., 2015**; **Trapp and Cork, 2015**). Additionally, PPG neurons in both the NTS and IRT express leptin receptors, respond to CCK, and project to the PVH. Nevertheless, our results suggest that GLP-1 innervation of PVH neurons, and the functions they regulate, are not uniformly specified by neonatal exposure to leptin, but that actions mediated by CRH may be particularly impacted by changes in GLP-1 innervation. Consistent with previous reports of altered stress profiles in mice with impaired leptin signaling (**Deck et al., 2017**), we observed an increase in the amount of time that LepRb$^{TB/TB}$ mice spent in the closed zone of an elevated plus maze, and a decrease in the number of transitions into a

different zone. However, re-expression of LepRb in PPG neurons did not restore normal plus maze performance, suggesting leptin signaling in PPG neurons alone is not sufficient to restore normal behavioral responses.

Changes in the density of GLP-1 inputs to the PVH in LepRb[TB/TB] mice phenocopied those of Lep[ob/ob] mice, suggesting LepRb signaling is required to alter the density of GLP-1 inputs to the PVH. Moreover, the developmental action of leptin on projections from PPG neurons to the PVH appears to be cell autonomous. When leptin receptor expression was restored specifically to PPG neurons, the density of GLP-1 inputs in LepRb[TB/TB] mice was normalized to wild-type levels, and transmission of viscerosensory activation of PVH neurons in response to i.p. CCK was normalized. Other developmental factors have been shown to act in a cell-autonomous manner. For example, mice with *Mecp2* mutations show severe neurological deficits, and increasing BDNF levels in these mice normalizes physiological and morphological phenotypes observed in the mutants through cell-autonomous signaling (*Sampathkumar et al., 2016*).

Although we found that nearly all PPG neurons respond to leptin during postnatal development, additional postsynaptic factors could play a role in cellular targeting of GLP-1 inputs to PVH neurons. For example, leptin could alter expression of a target-derived factor that specifies GLP-1 density onto particular cell types or onto diverse cell types in functionally distinct domains of the PVH. However, because our data demonstrate that leptin acts directly on PPG neurons to specify GLP-1 fiber density in the PVH, the action of a target-derived factor is probably not responsible for leptin's suppression of PVH innervation. An alternative mechanism is that synaptic space in the PVH of Lep[ob/ob] mice, vacated by the reduction in AgRP afferents, may become occupied by the elevated number of convergent GLP-1 inputs in these mice. A summary schematic of this model is shown in *Figure 8*. Whether leptin impacts GLP-1 projections to other terminal fields and cell types remains to be determined. However, AgRP and GLP-1 projections converge onto a number of forebrain nuclei where similar rearrangements of synaptic input to distinct subpopulations of neurons may occur.

Consistent with the neuroanatomical observation that leptin suppresses the density of projections from PPG neurons to the hypothalamus, we also determined that viscerosensory transmission from the gastrointestinal (GI) tract to the PVH is dysregulated. It is well-established that systemic administration of CCK activates subdiaphragmatic vagal mechanoreceptors and chemoreceptors and that vagal afferents convey this information from the gut to the NTS (*Richards et al., 1996*; *Rinaman, 1999*). In turn, hindbrain neurons that receive this viscerosensory information on GI nutritional status send direct projections to the PVH. Although hindbrain to hypothalamic circuitry is not required for the well-known hypophagic effects of exogenous CCK, the activation of PVH neuroendocrine neurons by CCK requires ascending projections arising from the hindbrain (*Maniscalco and Rinaman, 2013*). Thus, changes in levels of Fos labeling in the PVH observed in mice lacking robust leptin signaling are most likely due to alterations in the strength of these viscerosensory afferents.

Given the marked changes in neuroanatomical and neurophysiological phenotypes observed, it is surprising that re-expression of LepRb to PPG neurons was not sufficient to normalize food intake and anxiety-like behaviors of LepRb[TB/TB] mice. Several reports have implicated PPG neurons in the regulation of food intake and body weight by leptin. Knockdown of leptin receptors in the NTS leads to increased food intake and body weight gain (*Hayes et al., 2010*). In addition, deletion of leptin receptors fromPhox2b neurons, which include a subpopulation of PPG neurons in the NTS, resulted in increased food intake, increased body weight, and elevated metabolic rate (*Scott et al., 2011*). Similarly, chemogenetic activation of PPG neurons led to reduced food intake and metabolic rate (*Gaykema et al., 2017*), and depletion of PVH GLP-1R resulted in increased food intake and obesity (*Liu et al., 2017*). It remains unclear why targeted expression of LepRb in PPG neurons did not lead to normalization of behavioral parameters thought to be regulated by PPG neurons, that were found to be altered in LepRb[TB/TB] mice. It is possible that the obesity phenotype in LepRb[TB/TB] mice is so robust that physiological changes due to restored LepRb signaling in PPG neurons alone may be obscured in these mice. Consistent with this notion, Williams and colleagues found that administration of GLP-1 to obese leptin receptor-deficient Koletsky rats did not impact food intake (*Williams et al., 2006*). Additionally, compensatory adaptations may have occurred during development that resulted in no measurable effect on feeding behavior (*Luquet et al., 2005*). Leptin may be required for normal GLP-1 innervation to the PVH, but restoration of leptin signaling in PPG neurons alone may not be sufficient to normalize the marked metabolic dysfunction of LepRb[TB/TB] mice. Consistent with this interpretation, recent findings suggest that PPG neurons may not play a central role

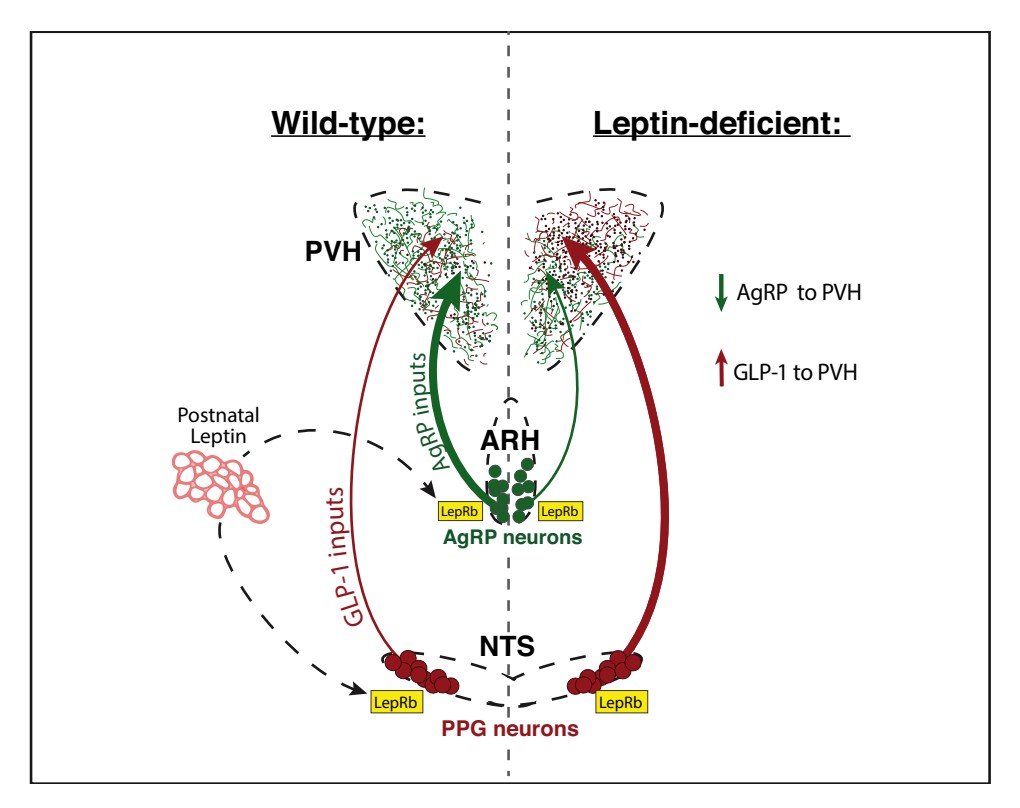

**Figure 8.** Leptin is required to specify the normal balance of hormonal and viscerosensory information conveyed by AgRP and GLP-1 inputs that converge onto the PVH. During development, the adipocyte-derived hormone leptin is secreted by adipose tissue and binds to leptin receptors (LepRb) located on AgRP neurons in the ARH, and on PPG neurons in the NTS. In animals with normal leptin signaling, the PVH contains a greater density of AgRP inputs compared with that of GLP-1. In the absence of leptin signaling, the density of AgRP inputs to the PVH decreases and those derived from PPG neurons in the NTS increase. Thus, in contrast to the growth-promoting actions of leptin on AgRP neurons, leptin appears to suppress innervation by PPG neurons to the PVH, through a receptor dependent and cell-autonomous mechanism. Leptin's developmental actions may function to balance excitatory and inhibitory inputs that converge onto multiple populations of PVH neurons and regulate their responsiveness to environmental signals.

in the normal regulation of food intake, but rather function to regulate state-dependent consummatory behavior within the context of anxiety and stress (*Holt et al., 2018*). PPG neurons appear to be activated under exceptional conditions, so that behavioral actions are aligned with avoidance of immediate danger. During imminent threat or perceived danger, behavioral priorities are reorganized and hypohagia is a secondary consequence. Unless an animal experiences substantial levels of stress or anxiety, PPG neurons may not achieve sufficient activation to produce changes in metabolic state. Therefore, uncovering the physiological impact of leptin's developmental effects on GLP-1 inputs to the PVH may require a comprehensive evaluation of how anxiety and stress interact to regulate ingestive behavior.

Although other neural systems regulating energy balance may compensate for perturbations in the density of PPG inputs to the PVH, it is clear that the cellular physiology of PVH GLP-1R expressing neurons is profoundly affected by the developmental actions of leptin. Consistent with the fact that the majority of PPG neurons are glutamatergic (*Card et al., 2018*), excitatory neurotransmission is enhanced in GLP-1R expressing neurons in Lep$^{ob/ob}$ mice, concomitant with elevated GLP-1 innervation. Moreover, the cell-autonomous developmental regulation of GLP-1 inputs to the PVH corresponds with altered neuronal activation in the PVH following stimulation of vagal afferents. Together these findings suggest that during development, leptin signals cell autonomously through its

receptor in the NTS to suppress outgrowth of PPG axons, thereby reducing excitatory inputs to PVH GLP-1R, resulting in reduced viscerosensory information being conveyed to PVH neurons.

# Materials and methods

## Key resources table

| Reagent type (species) or resource | Designation | Source or reference | Identifiers | Additional information |
|---|---|---|---|---|
| Genetic reagent (*M. musculus*) | tdTomato | Jackson Laboratory | Stock #:007914 RRID:IMSR_JAX:007914 | PMID:20023653 |
| Genetic reagent (*M. musculus*) | *Gcg*-Cre | PMID:28218622 | RRID:MMRRC_051056 | Dr. Michael Scott (University of Virginia) |
| Genetic reagent (*M. musculus*) | *LepRb-Cre* | Jackson Laboratory | Stock #:032457 RRID:IMSR_JAX:032457 | Dr. Martin Myers, Jr. (University of Michigan) PMID:17021368 |
| Genetic reagent (*M. musculus*) | $Lep^{ob/ob}$ | Jackson Laboratory | Stock #:000632 RRID:IMSR_JAX:000632 | $Lep^{ob/+}$ heterozygotes were bred to generate homozygous, leptin-deficient mice |
| Genetic reagent (*M. musculus*) | $LepR^{TB/TB}$ | Jackson Laboratory | Stock #:018989 RRID:IMSR_JAX:018989 | PMID:22326958 |
| Genetic reagent (*M. musculus*) | *Glp1r-Cre* | Jackson Laboratory | Stock #:029283 RRID:IMSR_JAX:029283 | PMID:27238020 |
| Genetic reagent (*M. musculus*) | Synaptophysin-tdTomato | Jackson Laboratory | Stock #:012570 RRID:IMSR_JAX:012570 | MGI ID: J:170755 |
| Antibody | Rabbit monoclonal anti-cFos | Cell Signaling | Cat. #: 2250S RRID:AB_2247211 | IHC (1:1000) |
| Antibody | Rabbit polyclonal anti-CRF | Dr. Paul Sawchenko; Dr. Wylie Vale (Salk Institute) | PBL#rC68 | IHC (1:1000) |
| Antibody | Mouse monoclonal anti-HuC/D | Molecular Probes | Cat. #: A21271 RRID:AB_221448 | IHC (1:500) |
| Antibody | Rabbit polyclonal anti-dsRed | Clontech | Cat. #: 632496 RRID:AB_10013483 | IHC (1:1000) |
| Antibody | Rabbit anti-Oxytocin | Peninsula | Cat. #: T-4084 RRID:AB_518524 | IHC (1:2500) |
| Antibody | Rabbit polyclonal anti-GLP-1 | Peninsula | Cat. #: T-4363 RRID:AB_518978 | IHC (1:5000) |
| Antibody | Rabbit polyclonal anti-pSTAT3 (Y705) | Cell Signaling | Cat. #: 9131S RRID:AB_331586 | IHC (1:1000) |
| Antibody | Goat polyclonal anti-rabbit Alexa Fluor 568 | ThermoFisher Scientific | Cat. #: A-11036 RRID:AB_10563566 | IHC (1:500) |
| Antibody | Goat polyclonal anti-rabbit Alexa Fluor 488 | ThermoFisher Scientific | Cat. #: A-11034 RRID:AB_2576217 | IHC (1:500) |
| Antibody | Goat polyclonal anti-mouse Alexa Fluor 647 | ThermoFisher Scientific | Cat. #: A-21236 RRID:AB_2535805 | IHC (1:500) |
| Recombinant DNA reagent | AAV pCAG-FLEX-EGFP-WPRE | Addgene | Cat. #: 51502 RRID:Addgene_51502 | PMID:24695228 |
| Peptide, recombinant protein | Leptin | Peprotech | Cat. #: 450–31 | 10 mg/kg i.p. |
| Peptide, recombinant protein | CCK | Bachem | Cat. #: H-2080 | 10 µg/kg i.p. |
| Commercial assay or kit | Corticosterone RIA | MP Biomedicals | Cat. #: 07–120102 | |

*Continued on next page*

*Continued*

| Reagent type (species) or resource | Designation | Source or reference | Identifiers | Additional information |
|---|---|---|---|---|
| Chemical compound, drug | Colchicine | Sigma | Cat. #: C9754 | 4 mg/mL |
| Software, algorithm | Imaris | Bitplane | V9.2 | |
| Software, algorithm | Volocity | PerkinElmer | V6.1.1 | |
| Software, algorithm | GraphPad Prism | Prism | Prism 7 | |

## Animals

Mice expressing Cre recombinase under control of the leptin receptor promoter (LepRb-Cre mice) were provided by Dr. Martin Myers, Jr., University of Michigan (*Leshan et al., 2006*). Transgenic BAC mice expressing Cre recombinase under control of the *Gcg* promoter (GCG-Cre mice) were generated at the University of Texas Southwestern and validated by Scott and colleagues (*Gaykema et al., 2017*). Mice expressing the Cre-dependent fluorescent reporters tdTomato (tdTom mice; Ai14D-Gt(Rosa)26Sor; stock number: 007914) and synaptophysin-tdTomato (SynTom mice; Ai34D-Rosa-CAG-LSL-Synaptophysin-tdTomato-WPRE; stock number: 012570) were obtained from The Jackson Laboratory. Knockin mice expressing an IRES-Cre fusion protein under control of the *Glp1r* promoter (GLP-1 R-Cre mice) were generated by Dr. Stephen Liberles, Harvard Medical School (*Williams et al., 2016*) and obtained from The Jackson Laboratory (stock number: 029283). Mice containing a Cre-dependent *Lox-P* flanked transcription blocker (loxTB) sequence between exons 16 and 17 of the leptin receptor gene (LepRb$^{TB/TB}$ mice) were obtained from The Jackson Laboratory (stock number: 018989) and validated by Elmquist and colleagues (*Berglund et al., 2012*). Experimental mice were generated through heterozygous intercrosses to generate homozygous, leptin-deficient offspring (*Lep$^{ob/ob}$* mice). WT littermates with normal leptin expression were used as controls. To visualize neurons in the NTS that express leptin receptors and neurons in the PVH that express GLP-1 receptors, LepRb-Cre mice and GLP-1 R-Cre mice were crossed with tdTom mice to generate LepRb-Cre::tdTom mice and GLP-1 R-Cre::tdTom mice. To visualize PPG inputs to the PVH, GCG-Cre mice were crossed with SynTom mice to generate GCG-Cre::SynTom mice. These mouse lines were then bred onto the leptin-deficient background to generate GLP-1 R-Cre::tdTom:: Lep$^{ob/ob}$ mice and GCG-Cre::SynTom:Lep$^{ob/ob}$ mice. WT controls were generated from the same litters.

All animal care and experimental procedures were performed in accordance with the guidelines of the National Institutes of Health and the Institutional Care and Use Committee of Vanderbilt University. Mice were housed at 22°C on a 12:12 hr light:dark cycle (lights on at 6:00 am:lights off at 6:00 pm). Mice were provided ad libitum access to a standard chow diet (PicoLab Rodent Diet 20 #5053). Mice were weaned at P22 and maintained with mixed genotype littermates until males were used for experiments.

## Immunohistochemistry and Treatments

### GLP-1 immunolabeling

WT and Lep$^{ob/ob}$ mice were perfused at P6, P10, P16, P24, and P60 days of age and processed for immunofluorescence by using an antibody to GLP-1 (1:5,000; Peninsula Laboratories, San Carlos, CA). Similarly, GCG-Cre::LepRb$^{TB/TB}$, LepRb$^{TB/TB}$ mice and WT control mice were also prepared for GLP-1 immunolabeling. Mice were first anesthetized with tribromoethanol (TBE) and then perfused transcardially with cold 0.9% saline, followed by cold fixative (4% paraformaldehyde in borate buffer, pH 9.5) for 20 min. Brains were then removed from the skull and postfixed in the same fixative for 4 hr. Brains were cryoprotected overnight in a 20% sucrose solution before being frozen in powdered dry ice and sectioned on a cryostat at 20 µm (neonatal brains; P6-P24), or at 30 µm (adults; P60) by using a sliding microtome. Brain sections were first mounted onto gelatin-subbed slides, rinsed in KPBS and then pretreated for 20 min in a 0.5% NaOH / H$_2$O$_2$ solution in KPBS and placed in 0.3% glycine for 10 min. Next, sections were incubated in 0.03% SDS, blocked in 4% normal goat serum containing 0.4% Triton-X and 1% BSA. The slide-mounted tissue sections were then incubated for 48 hr with a rabbit anti-GLP-1 antibody. Following primary antibody incubation, sections were rinsed

several times in 0.02M KPBS, incubated for 2 hr at room temperature in blocking buffer containing secondary antibodies against rabbit (raised in goat) conjugated with Alexa-Fluor fluorochromes (Life Technologies, Carlsbad, CA) and coverslipped using ProLong mounting medium (Life Technologies, Carlsbad, CA).

## Leptin activation

In order to determine if LepRb and PPG neurons in postnatal mice are responsive to leptin, LepRb-Cre::tdTom and GCG-Cre::SynTom mice received intraperitoneal (i.p.) injections of recombinant mouse leptin (10 mg/kg body weight; Peprotech Inc, Rocky Hill, NJ), or vehicle (0.9% sterile saline), at P16. Mice were anesthetized with TBE 45 min after leptin injection and perfused transcardially with 0.9% saline, followed by fixative (2% paraformaldehyde in phosphate buffer, pH 7.4) for 10 min. Brains were postfixed in the same fixative for 2 hr, and cryoprotected in 20% sucrose overnight. Each brain was sectioned at 20 μm using a cryostat and processed for pSTAT3 immunohistochemical labeling as described previously (*Bouret et al., 2012*). First, tissue sections were directly mounted onto superfrost slides. Next, the sections were pretreated in a 0.5% NaOH / $H_2O_2$ solution, then placed in 0.3% glycine. Sections were then incubated in 0.03% SDS, and blocked in a solution that contained 4% normal goat serum, 0.4% Triton-X, and 1% BSA. Tissue sections were incubated for 48 hr with a rabbit anti-pSTAT3 primary antibody (1:1,000; Cell Signaling, Danvers, MA). Following primary antibody incubation, sections were rinsed in 0.02M KPBS and incubated for 2 hr at room temperature in blocking buffer containing goat anti-rabbit Alexa-Fluor conjugated secondary antibodies (Life Technologies, Carlsbad, CA). Sections were coverslipped using ProLong antifade mounting medium (Life Technologies, Carlsbad, CA).

## Corticosterone measurements

Plasma corticosterone levels were measured using radioimmunoassay (MP Biomedicals, Santa Ana, CA). Blood was collected from the facial vein into chilled EDTA-coated tubes. Plasma was separated by centrifugation at 4°C 6000 rpm for 15 min and stored at −20°C.

## Postsynaptic target visualization

GLP-1 inputs to CRH and Oxytocin neurons in the PVH were visualized in GCG-Cre::SynTom::Lep[ob/ob] mice and GCG-Cre::SynTom::WT controls. To improve immunohistochemical labeling of CRH neurons, mice were first treated with colchicine. Colchicine (Sigma-Aldrich, Milwaukee, WI; 4 mg/mL in KPBS) was injected into the right lateral ventricle by using a glass micropipette. 24 hr after colchicine treatment, mice were perfused transcardially with the same fixative as described above for GLP-1 immunohistochemistry. Brain sections were incubated for 72 hr at 4°C in blocking buffer containing the following antisera: mouse anti-HuC/D (a pan-neuronal marker to identify the cytoarchitecture and borders of the PVH; 1:500; Life Technologies, Carlsbad, CA), and either a rabbit anti-CRH (PBL#rC68; generous gift from Drs. P. Sawchenko and W. Vale, Salk Institute, La Jolla, CA), or a rabbit anti-Oxytocin antiserum (1:2500; Peninsula Laboratories, San Carlos, CA). The primary antibodies were localized with corresponding Alexa Fluor conjugated secondary antibodies (Life Technologies). Sections were mounted onto gelatin-subbed slides and coverslipped with ProLong antifade mounting medium (Life Technologies, Carlsbad, CA).

## Activation of viscerosensory afferents to the NTS with systemic CCK

To test whether leptin alters the activity of PPG neurons and their downstream targets in response to a visceral stimulus, a previously validated Fos assay was adapted (*Maniscalco and Rinaman, 2013*). Systemic injections of CCK activate CCK-1 receptors on vagal afferents that in turn activate neurons in the caudal NTS, which project to the PVH and induce Fos expression (*Rinaman et al., 1994*). Accordingly, GCG-Cre::SynTom::Lep[ob/ob] mice and GCG-Cre::SynTom::WT controls were injected i.p. with CCK (10 μg/kg; Bachem H-2080, San Carlos, CA) or 0.9% sterile saline vehicle and sections through the NTS and PVH were processed for Fos immunohistochemistry. CCK was dissolved in sterile saline vehicle just before injection. This dose of CCK was utilized because it is known to activate ascending vagal afferents that project to the PVH (*Rinaman, 2003*). To minimize stress, mice were handled for up to a week prior to CCK administration. After injection with CCK or saline, mice were returned to their home cage and left undisturbed for 90 min. Mixed groups of

experimental and control mice were anesthetized and perfused as described above. Brain sections were incubated for 48 hr in a rabbit anti-Fos primary antibody (1:1000; Cell Signaling, Danvers, MA) which was localized by using Alexa Fluor conjugated secondary antibodies.

## Stereotaxic injections

LepRb-Cre mice were anesthetized with isoflurane (1.5–2.0%, 1 L/minute in $O_2$) and placed in a stereotaxic frame (Kopf Instruments, Tujunga, CA). The skull was exposed through a dorsal midline incision in the skin. The fourth ventricle and obex were identified and used as geographical landmarks to determine injection site in the NTS. The stereotaxic coordinates used were: A/P, −0.16; M/L, ±0.2; D/V, −0.2 from the obex. Viral injections were performed using glass micropipettes in a pressurized picospritzer system (General Valve Corporation, Fairfield, NJ; 40 pounds pressure per square inch; average duration 4msec). Then, 0.5 μL AAV pCAG-FLEX-EGFP-WPRE virus (Addgene, Cambridge, MA) was injected into the NTS. The tip of the glass micropipette was left in the brain for 5 min, and then slowly retracted. Mice were given analgesic and allowed to recover on a heating pad for 2 days post-surgery. They were then transcardially perfused as described above 14 days post-surgery.

## Image acquisition and analysis

Sections through the PVH with were examined on a laser scanning confocal microscope (Zeiss 710 or 800) and cytoarchitectonic features of the PVH, visualized with the cytoplasmic neuronal marker HuC/D (*Biag et al., 2012*) were used to define matching regions of interest (ROI) for quantitative analysis. Because the PVH contains functionally discrete subcompartments, we quantified the density of GLP-1 immunolabeled or GCG-Cre::SynTom-labeled fibers in the PVH in anatomically defined ROIs, as well as onto specific cell types. Thus, we quantified GLP-1 input density in the PVHmpd subcompartment, which is involved in the control of hormone secretion from the anterior pituitary, and in the PVHpml, because these cells innervate regions of the brainstem known to regulate autonomic neurons. Moreover, these subcompartments contain the highest density of labeled GLP-1 fibers, as reported previously in the rat (*Tauchi et al., 2008*). In addition to subnuclear quantification, we also measured the density of GLP-1 inputs onto specific cell types within in the PVH, as leptin has been shown to impact peptidergic innervation of neuroendocrine and sympathetic preautonomic neurons with both cell type and target-neuron specificity (*Bouyer and Simerly, 2013*). Because it was reported that PPG neurons primarily innervate CRH and Oxytocin neurons in the PVH (*Tauchi et al., 2008*; *Katsurada et al., 2014*; *Kanoski et al., 2016*), we measured inputs to CRH and Oxytocin neurons the PVH at higher magnification. Confocal image stacks were collected for each ROI through the entire thickness of the PVH at a frequency of 0.4 μm using the 40x objective and a frequency of 1.19 μm using a 20x objective. Velocity visualization software (PerkinElmer, Waltham, MA) was used to prepare 3D reconstructions of each multichannel set of images. To quantify the overall densities of labeled fibers in each ROI of the PVH, a previously validated methodology was utilized (*Bouyer and Simerly, 2013*). First, images were binarized, skeletonized, and the total fiber length was summed for each ROI throughout the image stack to obtain an estimate of the total density of GLP-1 fibers within that ROI.

The density of labeled inputs from PPG neurons onto identified CRH and Oxytocin neurons in the PVH was measured in GCG-Cre::SynTom::Lep$^{ob/ob}$ mice and GCG-Cre::SynTom::WT controls by using Imaris image analysis software (Bitplane V9.2, Salisbury Cove, Maine). Sections containing immunolabeled oxytocin, CRH, and genetically labeled GCG-Cre::SynTom fibers were segmented in each image stack by establishing an intensity threshold. Three-dimensional (3D) reconstructions of image volumes were then rendered for each multi-channel image stack using Imaris. Surface renderings of CRH or Oxytocin neurons were created and GCG-Cre::SynTom-labeled fibers and terminals were defined by using the Spots function. Spots identified as being in close apposition (voxel to voxel apposition) to the neuronal surface renderings were quantified. Voxels containing GCG-Cre::SynTom-labeled neuronal processes that did not make contact with the surface renderings of CRH or oxytocin neurons were excluded from analyses. Numbers of GCG-Cre::SynTom-labeled spots (terminals) were summed and divided by the total number of CRH or oxytocin neurons contained in each image stack to estimate the density of GCG-Cre::SynTom inputs onto each population of PVH neurons. The terms 'inputs' or 'terminals' used in this study refer to the visualization of

synaptophysin-tdTomato (SynTom) labeled axonal structures in close apposition to neuronal cell bodies that were labeled histochemically with CRH, Oxytocin, or the pan-neuronal cell marker HuC/D antibodies. The terms 'fibers', 'axons,' or 'projections' include, but are not limited to SynTom-labeled axon structures, but also GLP-1 immunohistochemical labeling, which can be detected in axon terminals as well as fibers *en passant*. Quantification of these axonal elements were only considered to be in close apposition onto a target neuron in the PVH if they were touching by voxel-to-voxel contact, with no unlabeled voxels between the presynaptic and postsynaptic elements of the presumptive synapse. Super-resolution optical microscopy, or ultrastructural visualization, will be necessary to directly measure synaptic contacts onto postsynaptic cells, but the density of genetically targeted and immunohistochemically labeled inputs to CRH and Oxytocin neurons appears to correlate with previous estimates of the density of synapses onto identified cell types within the PVH (*Bouyer and Simerly, 2013*).

To measure the density of Fos labeling in the NTS and PVH that resulted from i.p. injection of CCK, the number of Fos immunoreactive nuclei was counted in maximum projection confocal images through each region aided by Volocity software. Images of sections through the caudal NTS containing labeled GCG-Cre::SynTom neurons were matched using cytoarchitectonic features of the caudal brainstem such as the area postrema, large motor neurons of the DMX, and the central canal and used for quantification. The number of GCG-Cre::SynTom-expressing neurons in each section was counted manually and the number of Fos immunoreactive nuclei was counted using the object counting function in Volocity. The number of Fos immunoreactive nuclei that were colocalized to GCG-Cre::SynTom-expressing neurons in the NTS was expressed as a percentage of the total number of GCG-Cre::SynTom-expressing neurons counted within the caudal NTS. Numbers of Fos immunoreactive nuclei were also counted in the same ROIs through the PVH as described previously for GLP-1 fiber density measurements. Images containing the PVHmpd and PVHpml regions were identified and Volocity software was used to count the total number of Fos positive nuclei in the ROIs in the PVHpml and PVHmpd regions that receive dense GLP-1 inputs. Similarly, the number of pSTAT3 immunoreactive nuclei in LepRb- (LepRb-Cre::tdTom) and in PPG (GCG-Cre::SynTom) neurons was quantified in sections through the NTS by using object counting features of Volocity software and expressed as a percentage.

## Electrophysiology

In order to measure spontaneous synaptic currents, acute brain slices were prepared from GLP-1 R-Cre::tdTom::Lep$^{ob/ob}$ and GLP-1 R-Cre::tdTom::WT mice. Mice were anesthetized with isoflurane (5%) and perfused transcardially with ice-cold Choline-Cl slicing solution containing 105 mM Choline-Cl, 2.5 mM KCl, 1.25 mM NaH$_2$PO$_4$, 7 mM MgCl$_2$, 25 mM NaHCO$_3$, 25 mM Glucose, 11 mM Na-ascorbate, 3 mM Na-pyruvate, and 0.5 mM CaCl$_2$. Mice were decapitated, the brains rapidly removed from the skull and 300 µm-thick coronal sections cut into cold slicing solution on a Leica VT1200S vibratome. Slices were stored in oxygenated artificial cerebrospinal fluid (aCSF) and maintained at 32°C throughout the experiment. Whole-cell voltage clamp recordings (at a holding potential of −60 mV) were obtained from the cell bodies of fluorescently identified GLP-1 R-Cre::tdTom neurons. Miniature excitatory postsynaptic currents (mEPSCs) were recorded using a Multi-Clamp 700B amplifier. Signals were digitized through a Digidata 1550B, interfaced via pCLAMP 10 software (Molecular Devices, San Jose, CA). All recordings were performed at 32°C, maintained with a TC-344B temperature controller connected to an inline heater (64–0102) and heated PH-1 platform (Warner Instruments, Hamden, CT). Patch pipettes were pulled from borosilicate glass capillaries with resistances ranging from 3 to 5 MΩ when filled with pipette solution. The bath solution contained 124 mM NaCl, 2.5 mM KCl, 1 mM MgCl$_2$, 2 mM CaCl$_2$, 1.25 mM NaH$_2$PO$_4$, 25 mM glucose, 25 mM NaHCO$_3$, 1 mM Na-ascorbate and 1 mM Na-pyruvate. Postsynaptic AMPA-receptor mediated currents in response to spontaneous release of glutamate-containing vesicles from presynaptic terminals were recorded in the presence of the sodium channel blocker TTX (1 µM). The pipette solution contained 120 mM Cesium Methanesulfonate (CsMeSO$_4$), 5 mM CsCl, 3 mM Na-ascorbate, 4 mM MgCl$_2$, 10 mM HEPES, 2 mM ethylene glycol-bis-(aminoethyl ethane)-N,N,N',N'-tetraacetic acid (EGTA), 4 mM Na-ATP, 0.4 mM Na-GTP, 10 mM phosphocreatine, 0.5 mM CaCl$_2$, 5 mM glucose, 5 mM QX-314 pH 7.25 (CsOH). All signals were digitized at 20 kHz, filtered at 2 kHz, and analyzed offline with Clampfit software (Molecular Devices, San Jose, CA). Images of patched cells were

captured on a Zeiss Axioskop 2 FS Plus microscope equipped with a Ximea XiQ USB3 Vision camera with a 40x objective.

## Behavioral experiments

### Meal pattern testing

Feeding behavior was measured in 16×14×13 cm operant chambers (Med Associates). Mice were individually housed and adapted to the chambers one week before meal pattern testing. Mice lived in the operant chambers continuously for two weeks while spontaneous food intake measurements were recorded for 23 hr each day. At the start of each session per day, a fixed ratio (FR) reinforcement schedule was employed and mice needed to successfully press the lever in order to receive a 20 mg food pellet (Bio-Serv; Frenchtown, NJ). The balanced precision pellet diet used in meal pattern testing was comparable to standard chow, with a macronutrient composition of 22% protein, 66% carbohydrate, and 12% fat, with a caloric density of 3.62 kcal/g. Mice started at FR1 on the first day of training and the FR schedule was increased to FR5, FR10, FR20, until the mice reached FR30 and they remained on the FR30 schedule until the end of testing. The FR of 30 was chosen because this schedule was previously determined to minimize pellet waste and loss while reproducing free feeding intake patterns. This meal pattern paradigm was adapted from a previously validated protocol (*Richard et al., 2011*).

### Elevated zero maze

Mice were adapted to the empty test chamber (27.5 × 27.5 cm) for ten minutes before testing. Following acclimation, the mice are placed in the chamber and the total distance traveled, the percent of time in the closed zones, and the number of transitions between zones were recorded for up to 12 min.

### Novelty-suppressed feeding test

Mice received ad libitum access to their standard chow diet in their home cage for two hours each day during 4 days of training. Mice were food deprived overnight prior to the test day. The testing was performed in an open field chamber that measured 50 × 50 cm. One pellet of standard chow diet was placed in the center of the arena. Mice were tested individually, and each mouse was placed in the corner of the open field chamber and the total activity and latency to initiate food intake was recorded using animal tracking and recording software (ANYmaze; Stoelting Co, Wood Dale, IL) for 20 min. The total amount of food consumed was measured.

## Experimental design and statistical analyses

Group data are presented as mean values ± SEM. Statistical significance was determined using GraphPad Prism software. Student's t-test was used to compare data within two groups, using paired and unpaired tests where appropriate. One-way ANOVA followed by a pairwise post-hoc test was used to test for comparisons between three groups. p-values<0.05 were considered statistically significant.

## Acknowledgements

This work was supported by NIH grants R01DK106476 (RBS), F32DK108598 (JEB) and R01MH116694 (MMS). Behavioral experiments were performed in part through the use of the Murine Neurobehavior Core lab at the Vanderbilt University Medical Center. We thank Dr. Joel Elmquist for GCG-Cre mice. The authors thank Frohar Mirzai for genotyping assistance and Nicholas Thomas-Low for unparalleled histological assistance. We also wish to thank Dr. Amanda Elson for constructive feedback on previous drafts of the manuscript and technical assistance during early phases of this work.

## Additional information

### Funding

| Funder | Grant reference number | Author |
|---|---|---|
| National Institute of Diabetes and Digestive and Kidney Diseases | F32DK108598 | Jessica E Biddinger |
| National Institute of Diabetes and Digestive and Kidney Diseases | R01DK106476 | Richard Simerly |
| National Institute of Mental Health | R01MH116694 | Michael M Scott |

The funders had no role in study design, data collection and interpretation, or the decision to submit the work for publication.

### Author contributions

Jessica E Biddinger, Conceptualization, Resources, Data curation, Formal analysis, Funding acquisition, Validation, Investigation, Visualization, Methodology, Writing - original draft, Project administration, Writing - review and editing; Roman M Lazarenko, Investigation, Methodology; Michael M Scott, Resources, Validation, Visualization; Richard Simerly, Conceptualization, Supervision, Funding acquisition, Writing - original draft, Project administration, Writing - review and editing

### Author ORCIDs

Jessica E Biddinger  https://orcid.org/0000-0001-7718-4782
Roman M Lazarenko  https://orcid.org/0000-0003-0177-2961
Richard Simerly  https://orcid.org/0000-0001-5840-0152

### Ethics

Animal experimentation: All animal care and experimental procedures were performed in accordance with the guidelines of the National Institutes of Health and approved by the Institutional Care and Use Committee of Vanderbilt University (Protocol M1700113-00).

### Decision letter and Author response

Decision letter https://doi.org/10.7554/eLife.59857.sa1
Author response https://doi.org/10.7554/eLife.59857.sa2

## Additional files

### Supplementary files

· Transparent reporting form

### Data availability

All data generated are included in the manuscript.

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
