## [Decision Letter]

**Acceptance summary:**

Your findings will be of interest to both developmental neuroscientists and the field of obesity research.

**Decision letter after peer review:**

Thank you for submitting your article "Leptin suppresses development of GLP-1 Inputs to the Paraventricular Nucleus of the Hypothalamus" for consideration by *eLife*. Your article has been reviewed by three peer reviewers, one of whom is a member of our Board of Reviewing Editors, and the evaluation has been overseen by Catherine Dulac as the Senior Editor. The reviewers have opted to remain anonymous.

The reviewers have discussed the reviews with one another and the Reviewing Editor has drafted this decision to help you prepare a revised submission.

Summary:

Biddinger and colleagues describe results from a series of studies investigating the effects of leptin on the development of glucagon like peptide 1 neuronal innervation of the hypothalamus. They used multiple genetic models of leptin and leptin receptor deficiency and find that a lack of leptin signal causes increased GLP-1 innervation of the paraventricular nucleus of the hypothalamus (PVH). They also report that neurons expressing GLP-1 receptors have increased activity using patch recordings. Next, the authors also demonstrate that mice lacking leptin or leptin receptor (LepR) signaling have increased neuronal activity following administration of CCK and enhanced excitatory inputs to Glp1R-expressing PVH cells. Based on this evidence the authors suggest that leptin acts to suppress representation of excitatory afferents from PPG neurons and diminish the impact of visceral sensory input to the PVH. Somewhat curiously, reactivation of LepRb signaling in PPG neurons has no effect on the metabolic or behavioral effects accompanying leptin signaling deficiency. Collectively, the studies are well described and provide novel findings that will be of wide interest. The use of complementary mouse models (ob/ob and LepRb TB/TB) and projection tracing techniques strengthens the observations made and make clear that leptin plays a role in the development of PPG inputs to the PVH. The finding that leptin acts to suppress development of PPG inputs to the PVH is novel and will be of interest to developmental neurobiologists.

Essential revisions:

The major issue is that the physiological relevance of these observations is not clear. Though reactivation of LepRb signaling in PPG neurons restores control levels of PVH innervation, there is no change in the energy balance or behavioral abnormalities associated with absence of leptin or leptin signaling. Moreover, given the authors emphasis on demonstrating changes in PPG innervation of CRH neurons in the absence of leptin signaling, the lack of any effect on corticosterone or anxiety-like behaviors is somewhat perplexing. The authors do show that reactivation of LepRb signaling in PPG neurons restores the exaggerated Fos response to CCK injection to wildtype levels, but again the physiologic relevance of this is not explored. At a minimum this needs to be discussed in detail.

Similarly, the changes observed in the current study appear at the extremes of leptin signaling or expression. That is the authors used leptin deficient and leptin receptor null reactivation mouse models in order to see the altered GLP-1 inputs to the PVH. Are these effects expected to be observed in the context of moderate changes in circulating leptin levels?

Revisions expected in follow-up work:

Regarding the electrophysiology, expanded traces would better illustrate an EPSC that are represented by the fast transients. This could be accomplished by expanding the trace to show the fast rise time and exponential decay of the events.

Given the demonstrated importance of the timing of leptin action in the development of arcuate hypothalamic to PVH projections, is the impact of leptin on PPG to PVH neural circuitry confined to discrete developmental window? Does leptin administration to ob/ob mice in early life restore normal PPG inputs to the PVH? Placing the "pruning" actions of leptin in the developmental context of leptin's effects on axon outgrowth would benefit our understanding of leptin dependent neural development.

The suppression of GLP-1 inputs to the PVH does not appear uniform. It might be worth noting whether a quantifiable decrease in GLP-1 terminals in the PVH as a whole is observed versus the need for sub-nuclei quantification.

Similarly, as noted by the authors, PPG neurons project to multiple brain areas. Unfortunately, none of these other projection areas are examined for leptin-dependent alterations in PPG density. This is important as it would provide insight regarding whether or not the effects of leptin are projection site specific or impact all PPG neurons.

Beyond the observation that leptin/leptin signaling in PPG neurons suppress PVH inputs, there is limited molecular insight into the underlying mechanisms at play in the "pruning" actions of leptin. The authors mention that PNMT fibers to the PVH decrease from P1 levels to levels seen in adulthood. Are PNMT fibers to the PVH altered by the absence of leptin? Can the absence of leptin alter pruning of neurons that do NOT express leptin receptors?

---

## [Author Response]

Essential revisions:The major issue is that the physiological relevance of these observations is not clear. Though reactivation of LepRb signaling in PPG neurons restores control levels of PVH innervation, there is no change in the energy balance or behavioral abnormalities associated with absence of leptin or leptin signaling. Moreover, given the authors emphasis on demonstrating changes in PPG innervation of CRH neurons in the absence of leptin signaling, the lack of any effect on corticosterone or anxiety-like behaviors is somewhat perplexing. The authors do show that reactivation of LepRb signaling in PPG neurons restores the exaggerated Fos response to CCK injection to wildtype levels, but again the physiologic relevance of this is not explored. At a minimum this needs to be discussed in detail.

Like the reviewers, we were surprised by the inability of LepRb restoration, in GCG-Cre::LepRTB/TB mice with leptin receptors expressed only on PPG neurons, to normalize the impaired physiological outcomes in LepRTB/TB mice and yet the GLP-1 innervation is rescued in the PVH. Given the demonstrated importance of GLP-1 input to the PVH it seems unlikely that rescuing the GLP-1 innervation would be of little physiological consequence. Instead, our finding suggests that in the LepRTB/TB model, leptin action on PPG neurons in the NTS is required but not sufficient to produce physiological or behavioral outcomes. Consistent with this interpretation, recent studies on the function of PPG neurons determined that a role for any metabolic effects of PPG neurons is not necessary under “normal” conditions, but that PPG neurons are recruited when challenged with stronger physiological stimuli (Holt et al., 2019). Additionally, the leptin-deficient phenotypes are so severe that they may mask a more subtle effect on physiological regulation. We have updated the Discussion to stress this issue and provide more clarity.

Similarly, the changes observed in the current study appear at the extremes of leptin signaling or expression. That is the authors used leptin deficient and leptin receptor null reactivation mouse models in order to see the altered GLP-1 inputs to the PVH. Are these effects expected to be observed in the context of moderate changes in circulating leptin levels?

The experiments in the current study do not allow us to determine how moderate leptin levels impact GLP-1 inputs to the PVH. It would be interesting to determine if leptin influences development of GLP-1 projections in a dose-dependent manner. In a separate project currently ongoing in the laboratory, we increased leptin levels during development and also observed effects on GLP-1 innervation, in the opposite direction than those reported here. Although more experiments are necessary, this suggests leptin may “tune” GLP-1 inputs to the PVH, and that optimum leptin levels are necessary for proper GLP-1 axon outgrowth.

Revisions expected in follow-up work:Regarding the electrophysiology, expanded traces would better illustrate an EPSC that are represented by the fast transients. This could be accomplished by expanding the trace to show the fast rise time and exponential decay of the events.

Expanded representative traces have been incorporated into Figure 5 of the revised manuscript to better display the rise and decay of mEPSCs. Additionally, we included data demonstrating leptin does not impact the kinetics of the events, and we added averaged traces normalized to peak current that illustrate that there is no significant difference in rise or decay time between wild-type and Lep^ob/ob^ mice.

Given the demonstrated importance of the timing of leptin action in the development of arcuate hypothalamic to PVH projections, is the impact of leptin on PPG to PVH neural circuitry confined to discrete developmental window? Does leptin administration to ob/ob mice in early life restore normal PPG inputs to the PVH? Placing the "pruning" actions of leptin in the developmental context of leptin's effects on axon outgrowth would benefit our understanding of leptin dependent neural development.

These are interesting questions and suggestions. We agree that these issues should be addressed. One approach would be to give daily injections of leptin during discrete postnatal periods. Such a study may be beyond the scope of the current study and the logistics are quite difficult during the altered work environment imposed by COVID-19. Moreover, a better strategy may be to take a genetic approach. In a collaboration with collaborators at Columbia University, we plan to utilize a conditional, inducible leptin mouse model to examine whether leptin exposure during critical periods of development impacts GLP-1 neuronal projections and metabolic physiology.

The suppression of GLP-1 inputs to the PVH does not appear uniform. It might be worth noting whether a quantifiable decrease in GLP-1 terminals in the PVH as a whole is observed versus the need for sub-nuclei quantification.

As noted by the reviewers, the suppression of PPG inputs to the PVH by leptin is not uniform. The PVH contains functionally discrete compartments that correspond to neuroanatomical features of the nucleus. As demonstrated here, leptin impacts innervation of PVH subcompartments with both cell-type and target-neuron specificity. Since GLP-1 primarily innervates parvicellular neuroendocrine neurons, notably CRH and Oxytocin neurons in the PVH, we chose to focus on these populations. We have added a more detailed rationale for this analysis to the Materials and methods section.

Similarly, as noted by the authors, PPG neurons project to multiple brain areas. Unfortunately, none of these other projection areas are examined for leptin-dependent alterations in PPG density. This is important as it would provide insight regarding whether or not the effects of leptin are projection site specific or impact all PPG neurons.

As noted by the reviewers, PPG neurons innervate multiple brain regions. The arcuate nucleus of the hypothalamus (ARH) is implicated in metabolic homeostasis, contains a high density of leptin receptors, and is densely innervated by PPG neurons. Therefore, in the revised manuscript we have included images and quantification of GCG-Cre::SynTom labeling in WT and leptin-deficient mice in the ARH (Figure 3). In contrast to its effects on GLP-1 inputs to the PVH, we did not see a significant change in ARH innervation, suggesting that leptin may influence the distribution of GLP-1 inputs in a target-specific manner. Based on these results, we plan to pursue a separate and more comprehensive study on the effects of leptin on development of GLP-1 inputs to all of its major terminal fields.

Beyond the observation that leptin/leptin signaling in PPG neurons suppress PVH inputs, there is limited molecular insight into the underlying mechanisms at play in the "pruning" actions of leptin. The authors mention that PNMT fibers to the PVH decrease from P1 levels to levels seen in adulthood. Are PNMT fibers to the PVH altered by the absence of leptin? Can the absence of leptin alter pruning of neurons that do NOT express leptin receptors?

This is a great point. At this time, we do not know if leptin impacts the development of PNMT fibers to the PVH. Several studies have shown that catecholamine neurons in the NTS do not express leptin receptors. However, this does not exclude a role for leptin during development on catecholaminergic projections from the hindbrain to the PVH. The other issue raised about whether leptin can impact circuit architecture through mechanisms that are independent of leptin receptors is currently being investigated in an unrelated study.